# Frugal engineering of a jaw crusher using the factor-of-frugality, a modern version of the safety-factor

**Aditya Krishnan**[iD], **Balkrishna C. Rao**[iD]*

Engineering Design Department, Indian Institute of Technology, Chennai, India

* balkrish@iitm.ac.in

## Abstract

Businesses are increasingly keen on going frugal due to increasing demand for sustainable and low-cost products that do not sacrifice quality. However, there is a dearth of tools for the systematic design and engineering of frugal products from scratch in the industry. Accordingly, a new approach has been applied in this paper for the formal design of a frugal jaw crusher for the mining industry. Consequently, this paper uses the *factor of frugality* (F of $F^N$) which is a composite number that combines the *safety factor* (N) with fractions of material saved in various stages of product-development. In doing so, this work has iteratively applied the *factor of frugality* to the relevant components of a jaw crusher. And rigorous design procedures are adopted, for maintaining quality, due to the use of lower *safety factors* in making the product frugal. Contemporary concepts like generative design, *design for manufacturing* and biomimetics have been explored to achieve frugality in the relevant "bulky" components of a jaw crusher. Accordingly, *factors of frugality* of $1.87^{1.79}$, $1.76^{1.68}$ and $1.63^{1.33}$ have been obtained for the flywheel, the Pitman and the rear wall, respectively, based on the frugal approach. Therefore, use of the new *frugal design approach* has resulted in material savings of 8%, 7% and 30% and increases in *factor of safety* of 35%, 71% and 18% in the flywheel, the Pitman and the rear wall, respectively, over their base values. Such savings in materials have been accompanied by moderate cost reductions with improvement in functionality and this trio of features is typical of an *advanced frugal product*.

## Introduction

In recent years, frugal products consuming lesser resources have appeared in a range of sectors. While frugal products are generally considered to be a grassroots phenomenon [1–3,28], the concept of engineered frugality is perhaps best explained by Rao, [2,3,28] Zeschy et al. [4] and Rosca and Bendul, [5] who suggest that frugal products can be developed by industry through introduction of measurable

**Data availability statement:** All the data necessary are given in the manuscript.

**Funding:** The author(s) received no specific funding for this work.

**Competing interests:** The authors have declared that no competing interests exist.

constraints on resources in the product development process. These constraints are primarily exercised on raw materials and costs which in turn influence different aspects of product development process including manufacturing, supply chains etc. Consequently, an engineered frugal product differs from an improvised or makeshift one with the former being identified as an Advanced Frugal Innovation (AFI) [2,3,28]. Moreover, the constraints that lend themselves to designing AFIs are also influential in designing an environmentally sustainable product [6,7]. Therefore, frugal products and in particular AFIs, are sustainable by nature [7–12]. This is because AFIs are defined to consume minimal amounts of resources while giving good functionality affordably. Minimal consumption of resources will typically have a beneficial effect on the environment and the low-cost with good functionality can uplift living standards. Both the positive environmental impact and affordable quality of AFIs sit in well with the tenets of the *sustainable development goals* (SDGs) [13].

The adoption of frugality in design and manufacturing is key to efforts of industries in rich and emerging markets for complying with environmental norms while also competing on cost, demand and quality [10,14]. There are several methodologies that can be used to frugally engineer a product. The approach called *design for manufacture and assembly* (DFMA) by Boothroyd and Dewhurst [15] is focused on reducing assembly complexity while also making components easier to manufacture by following well-established design principles. Another related work pertains to measurement of frugality by Kumar and Agarwal [16] where their quantification uses a *frugal manufacturing index* based on lean principles. Although this effort is exhaustive and covers some product development aspects, it excludes design that is the most crucial step for generating frugality. Other approaches including *design for recycling* and *design for quality* are also geared to frugal engineering of a product in that any methodology that reduces resource consumption also promotes frugality. In fact, Micaëlli et al. [17] propose use of the term *design for frugality* (DFF) to cover any methodology that promotes frugality. Hitherto, the quantification of frugality has been carried out by measuring and comparing various design alternatives using cost. While low cost is undoubtedly a primary attribute of a frugal product, the exclusion of resource-consumption and quality [17] would significantly hinder creation of a truly advanced frugal product. Against this backdrop, a more general approach has been detailed by Rao [12] that uses a three-step methodology to systematically engineer an *advanced frugal product* from scratch. Rao's approach [12] uses the *factor of frugality* in designing an AFI by accounting for every stage of product development [3,18,19,28] The *factor of frugality* is a composite number based on the *factor of safety* and the fractions of material saved in the various stages of product development. In particular, the *safety factor* is fixed at the lowest feasible value for maximum material-savings, while searching for yet more savings in materials at all stages of product development. This design approach considers direct material savings – and hence proportionate cost savings – as the primary objective for frugalization rather than reducing costs through other avenues such as labor and supply chains [3]. The traditional approach of designing products with a high *factor of safety* is eschewed in favor of well-engineered products where the level of uncertainty is minimized and

material savings are maximized. In other words, the *factor of frugality* is based on executing rigorous design procedures with the best of theoretical models and accurate data to compensate for uncertainties arising from lower *safety factors* [3,18,19].

Therefore, this paper addresses the novelty of applying frugal engineering principles to a jaw crusher by focusing on:

1. Frugal design is quantified using the *factor of frugality*. This allows the quantification of frugality in any product, including the jaw crusher, using a metric that hews to the three features of frugal engineering, i.e., low-cost, low-resource, and better quality.

2. The impact of the *factor of frugality* based design has been studied using the metrics of performance and cost.

3. The *factor of frugality* has been formulated for design of individual components of a given product. Consequently, a method of determining the overall *factor of frugality* of an assembly based on this formulation has been introduced in this effort.

4. This paper is the first instance of the methodical application of the *factor of frugality* to an actual industrial product. Accordingly, this paper presents the different versions of the product resulting from the iterative application of the *factor of frugality* while converging onto the final design. The advantages and limitations of the frugal methodology will become apparent in the process of designing this industrially relevant product.

## Methods

### Selection of a suitable industry level product

The *factor of frugality* approach can be applied to any product in any sector, irrespective of safety requirements since it is based on rigorous design procedures. In other words, safety is never sacrificed in arriving at a *frugal* product by employing rigorous design procedures using current data and models that are accurate with minimal uncertainty [18] Accordingly, a jaw crusher, a typical resource-consuming product from the mining industry has been selected to bring out the efficacy of this approach. Therefore, this paper will focus on redesigning relevant "bulky" components from this product using the *factor of frugality* approach.

A jaw crusher is an equipment used in the mining industry for primary crushing of minerals. It works by breaking rocks under compression and impact between a fixed- and a moving-jaw. A Blake-type single toggle jaw crusher [20] has been selected in this paper whose major components have been illustrated in Fig 1.

For this particular study, the components from the moving jaw assembly were considered, see Table 1. Of these, the top three heaviest components were taken up for redesign. The components are: Pitman frame, Rear wall and Flywheel. Referring to Table 1, the weights have been obtained from the initial designs of these components, based on data from *computer aided design* (CAD) models. The initial designs were created by adopting dimensional and technical data available on manufacturers' websites [21–26]. The final specifications of the machine were arrived at by using the methods outlined by Gupta and Yan [20].

### Use of the factor of frugality

The *factor of frugality* is a new metric for design and, engineering in general, that achieves best functionality under resource-and-cost constraints by subsuming the *classical safety factor* while also focusing on individual stages in *product development.* It is a modern version of the *factor of safety* that focuses on safety, resource, cost and quality in design.

The *factor of frugality* approach is initiated by fixing N at a low value of 1.5, as seen in Fig 2 [3]. The low value of N is arbitrary in that it should be lowest possible value commensurate with the current body of knowledge in a given area of engineering. Subsequently, the material saving schemes pertaining to stages of product development, i.e., design & materials, manufacturing and salvaging, are applied at this constant value of N.

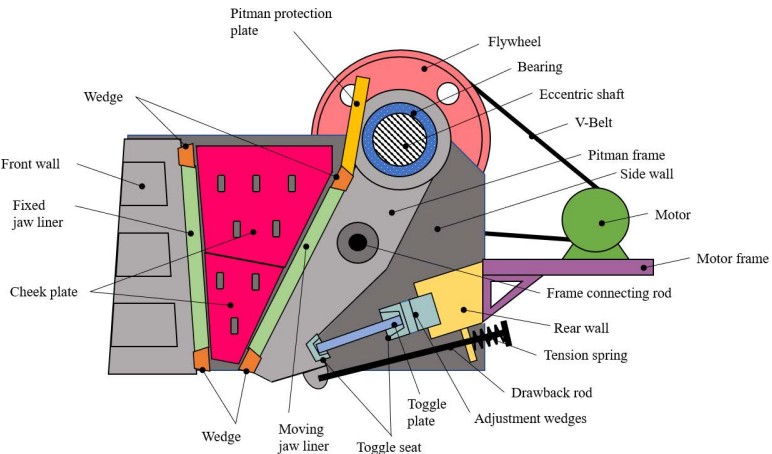

**Fig 1. Major components of a jaw crusher.**

**Table 1. List of components in moving jaw assembly.**

| Name of the component | Mass *(Kg)* | Mfg. method | Qty. |
|---|---|---|---|
| Pitman frame | 2932 | Casting | 1 |
| Rear wall | 1209 | Casting | 1 |
| Shaft | 1098 | Forging | 1 |
| Wedge assembly | 834 | Casting | 1 |
| Liner | 826 | Casting | 1 |
| Flywheel | 627 | Casting | 2 |
| Motor frame | 504 | Fabrication | 1 |
| Flange | 391 | Casting | 2 |
| Toggle plate | 258 | Casting | 1 |
| Bearing | 112 | Various | 2 |
| Pitman protection plate | 107 | Casting | 1 |
| Other components | 477 | Various | – |

Fig 2 brings out the working of the *factor of* frugality, i.e., F of $F^N$, for a shaft which is a workhorse of many engineering applications. As seen in Fig 2, a low value of N, at 1.5, is fixed throughout the *frugal product development* process. Such a low value of N generally results in lower material consumption and subsequent stages of *product development* are examined for more material savings in addition to that for this low value of N. Consequently, the *material saved* (MS) parameters account for weight of material saved in the design, materials and manufacturing stages with focus on salvaging or recycling. Therefore, low-resource consumption and, hence lower costs, are made possible from both low N and individual stages of *product development.* The quality is maintained at the highest level by hewing to the rigor of most accurate design and engineering principles.

However, some changes with this idealized setting of the *factor of frugality* are in order. It should be noted that although N is ideally fixed, it will be observed to vary slightly between various versions of a real time jaw crusher studied in this work. Even small changes in design may add to the complexity of the overall shape but can have a significant impact on material cost [18,27]. It is therefore important to design structures keeping in mind its manufacturability to avoid escalation of cost. Structural features can be optimized and redesigned to reduce resource utilization without compromising

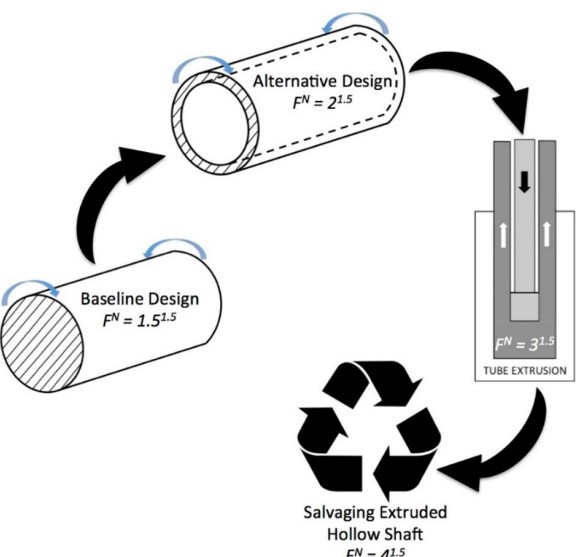

**Fig 2. Working of the factor of frugality.** Source: Rao B.C.,2019.

functionality [7]. The baseline for a given problem will typically be an existing "bulky" design that is commonly available. A systematic approach [3] is undertaken wherein the baseline is put through an iterative process of improvement using the *factor of frugality* which is computed for each component at every iteration, as shown in Fig 3. The notation for the *factor of frugality* is $F^N$ where F is calculated as:

$$F = N + \sum_{i=1}^{5} MS_i$$

(1)

where,

 N = the *factor of safety*

$MS_i$ = the fraction of material saved using the $i^{th}$ material savings scheme, see Fig 3

 As shown in Fig 3, equation (1) is iteratively applied for creating successive versions of a design, that improve upon the baseline by improving F, to arrive at an *advanced frugal product* [3]. More details on the theory and application of the *factor of frugality* ($F^N$) can be found in Rao's works [18,19,28,29].

 The *frugal design approach* (FDA) outlined above saves material through various schemes including manufacturing. In this regard, *design for manufacturing* (DFM) is used as an important tool in each iteration with $F^N$ to make the design suitable for manufacturing and exploit the material-properties for creating resilient and robust structures [30,31]. Use is made here of the principles of *frugal manufacturing* which aims for ideally zero waste from a low-cost single-pass process that produces products with excellent quality [19]. This goes well with considerations of sustainability where better functionality, lower costs, reduction of material consumption, waste reduction and energy reduction are important goals [32].

 The FDA also incorporates modern design concepts like biomimetics and generative design to achieve the best results. Generative design is used in this work since it tends to create natural-looking structures due to its evolutionary working principle [33]. Biomimetics and hence, bio-inspiration has been linked to sustainability and reported to produce innovative and frugal products [18,33–36]. The effect of such techniques on the frugality of the product design is studied, compared and contrasted using $F^N$ in the sections to follow.

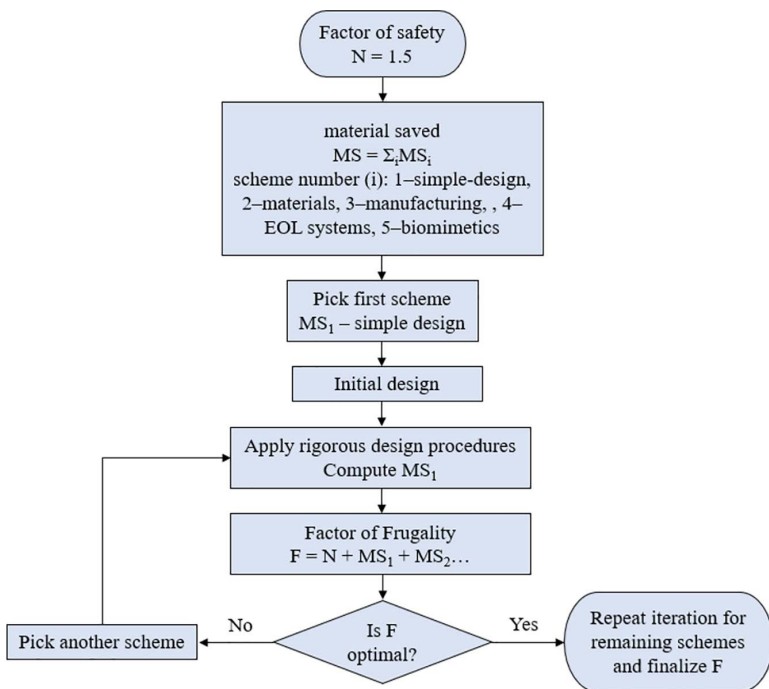

**Fig 3. Iterative frugal design process.** Source: Rao B.C., 2019.

## Numerical modeling

The development of 3D models of components and their subsequent stress analysis were carried out using a CAD software, Autodesk® Inventor®. A linear static analysis for studying stress distributions was implemented in all cases. The materials were simulated within their Hookean domain since plastic flow is detrimental to product-functionality. The load inputs for the analyses were calculated using previously established methods and formulae [37]. The loads and constraints were then identically applied to all versions of a given component. The details of the finite element analysis are tabulated in Table 2.

## Results

The crusher is designed for a power of *110 kW* at a shaft-speed of *230 RPM* with a feed acceptance of *600 mm* [20–26]. The material to be crushed is granite. Some basic assumptions to calculate the forces on each component using this data are:

**Table 2. General mesh and convergence details for stress analysis.**

| Component | Flywheel | Pitman frame | Rear wall |
|---|---|---|---|
| Mesh average element size *(mm)* | 14.3 | 22.8 | 13.5 |
| Minimum element size *(mm)* | 2.86 | 4.56 | 2.7 |
| Mesh type | 10-Node Tetrahedral | | |
| Grading Factor | 1.5 | | |
| Maximum turn angle | 60° | | |
| Convergence criterion (Von Mises stress) | 5% | | |
| h-refinement threshold/ No. of refinements | 0.75/ 3 | | |

- The incoming rock geometry is assumed to be cubical with a side of *600 mm*.

- The rock is assumed to be loaded in a 3-point bending configuration due to the corrugated nature of the liner plates.

- The modulus of rupture of granite is approximately *25 MPa*. [38]

Using the formula for Modulus of Rupture [39] ($\sigma$) given by equation (2), the force required to break the rock is calculated.

$$\sigma = \frac{3PL}{2bd^2}$$

(2)

where,

 P is the necessary breaking-force or load applied
 L is the length of the specimen
 B is the breadth of the specimen
 D is the depth of the specimen

For *L=b=d=0.6 m* and a σ of *25 MPa* in equation (2), the force *(P)* required to break the rock is determined to be *$6 \times 10^6$ N*. This value when used with the equations for a single toggle jaw crusher [37] results in a torque of *170586 N-m* for breaking the rock. Of this total torque, *87577 N-m* is delivered from the driven-flywheel which is connected to the motor.

 The toggle plate is designed to fail at a force of *$1.5 \times 10^6$ N*, which according to Eqn (2), corresponds to a cubic rock of side 0.3 m being broken at the lower portion of the crusher, assuming the rock has fallen to about 75% of the length of the pitman liner.

 The toggle's failure load is taken as the loading criterion for the pitman as well as the rear wall. Equations for a single toggle jaw crusher were again used to calculate the load applied on the face of the Pitman frame. This value was found to be *$1.77 \times 10^6$ N*. The torque and load values were used in the *finite element analyses* (FEA) of relevant components.

## Flywheel

A flywheel uses inertia to store energy through its rotation. In the context of a jaw crusher, it is used to provide the extra energy required to crush rocks during the crushing stroke. A frugal design of a flywheel is challenging as a flywheel needs mass to increase its rotational *moment of inertia* (I). Hence, the component must be designed such that the mass is reduced while keeping the *moment of inertia* unchanged. This entails moving mass that is near the hub of the flywheel to the periphery while increasing the strength of the design. The flywheel is to be designed for a peak torque of *87577 N-m* and a belt tension of *14.1 kN*, which are cyclical loads. The required *moment of inertia* of the flywheel is fixed at *183 Kg-m²,* calculated from the torque. The material chosen for casting the flywheel is ASTM A48, as it is known to have good shock-absorption properties. The 3D models for each version are put through stress analysis in Autodesk® Inventor® under the same loading conditions and constraints.

 Version 1, shown in Fig 4, is a simple design of flywheel featuring a flat profile for ease of manufacturing. Stress analysis shown in Fig 5 reveals that there is stress concentration at the base of each spoke. The torque applied at the periphery is transferred to the hub via the spokes. As the distance to the hub decreases, there is an increase in the bending moment and subsequently the stress value. As seen in Table 3, this version of flywheel-design had the highest peak stress value and the lowest endurance strength of the three designs that were assessed.

 Version 2, shown in Fig 4, is a result of generative design. Symmetry constraints were applied on mutually perpendicular planes passing through the axis of the flywheel, while preserving the rim and hub geometry to run generative design. A rounded rectangular cross section with width to height ratio of 2:1 was adopted to achieve lower stresses considering the fact that arms of the flywheel predominantly undergo bending. The tapered profile of the arms is also fashioned like beams of uniform strength [40]. Subsequent stress analysis reveals that there is a concentration of stress at the base of

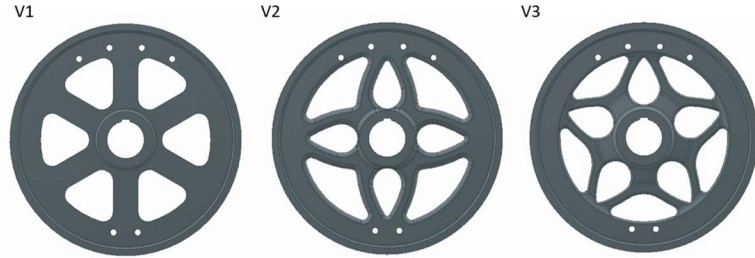

**Fig 4. Comparison of Versions 1, 2 and 3 of flywheel design.**

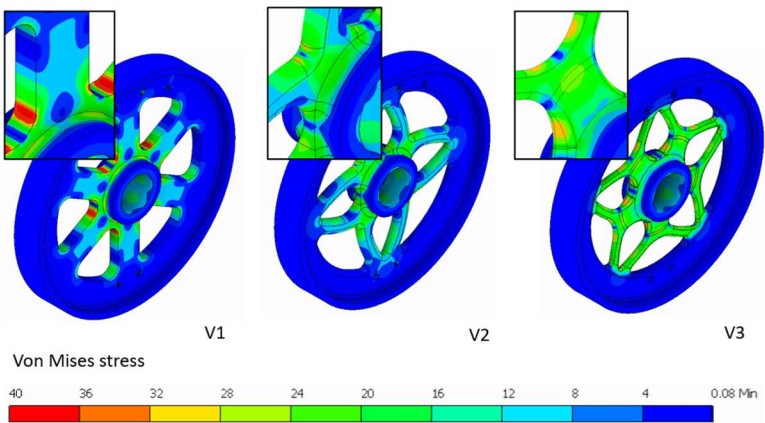

**Fig 5. Stress analysis of flywheel.** The insets zoom in on the spoke base and hub.

**Table 3. Summary of results for flywheel design.**

| Version | 1 | 2 | 3 |
|---|---|---|---|
| Mass *(Kg)* | 627 | 597 | 578 |
| Endurance Strength *(MPa)* | 49.05 | 50.3 | 54.71 |
| Peak Stress *(MPa)* | 37.1 | 30.2 | 30.6 |
| N (Peak stress/Endurance strength) | 1.32 | 1.67 | 1.79 |
| ΣMS | 0 | 0.05 | 0.08 |
| F | 1.32 | 1.72 | 1.87 |
| $F^N$ | $1.32^{1.32}$ | $1.72^{1.67}$ | $1.87^{1.79}$ |

each spoke like Version 1 but significantly reduced in magnitude. The stress contour plot in Fig 5 also shows that there are areas of very low stress where more material can be removed. Since Version 2 uses only a simple design scheme to reduce mass, $MS_1 = 0.05$ and $\Sigma MS = 0.05$, as shown in Table 3.

Version 3, shown in Fig 4, is an improvement over Version 2 where the same petalled structure is maintained but distinct branching, seen in trees, has been introduced. The cross section of the spokes, similar to Version 2, is a rounded rectangle with a width to height ratio of 2:1, but with a smaller cross-section. The cross section of the trunk increases closer to the hub. This particular arrangement has been shown to improve the damping and load bearing properties in trees [41,42]. An increase in branching tends to reduce overall stress and improves damping properties. The odd number

of spokes and the curved design, as per recommendation in [31], lend themselves to reduced residual stresses during manufacturing. A five-spoke-arrangement was chosen, as it is a common shape found in nature, that improves upon the previously designed four-spoke structure, distributing the stress more evenly, as shown in Fig 5. As per the data in Table 3, there is also a significant increase in the value of endurance strength, owing to the reduced section thickness. Stress analysis reveals that there is a reduction in overall stress and a more uniform distribution in the component, as seen in Fig 5. Since Version 3 uses the same design principles as Version 2, $MS_1 = 0.05$. Additionally, biomimetics contributes to more material savings, giving $MS_5 = 0.03$. Therefore, $\Sigma MS = 0.05 + 0.03 = 0.08$, as shown in Table 3.

It should be noted that N in all these cases is the fatigue-based value. It is the ratio of peak stress to endurance strength. Endurance changes with change in thickness and is calculated using standard procedure [43]. A summary of results and calculated values of F and $F^N$ for all versions can be seen in Table 3.

## Pitman frame

The pitman frame performs the vital function of providing mounting surfaces for the crusher's mechanism. As a result, the Pitman frame is subjected to high levels of loading. Accordingly, the material selected is ASTM A352 (alloy steel). To protect the Pitman, a toggle plate acts as a fuse during load spikes and breaks before any damage occurs to the Pitman. Hence, the design load of the toggle plate is taken as an input for the stress analysis of the Pitman frame, see Fig 6. The 3D model for each version is analyzed for stresses and their distribution through Autodesk® Inventor® with loading and constraints kept constant for each version.

Version 1, shown in Fig 7, is a simple design satisfying assembly and packaging constraints. Basic manufacturing feasibility has been accounted for, but an in-depth DFM was not performed on this initial version. The stress analysis, shown in Fig 8, reveals that the stress is concentrated near the base of the casting where the section is inadequate or improperly designed. Furthermore, the ribs at the rear also carry a significant amount of stress that could be attributed to the overall geometry.

Version 2, shown in Fig 7, incorporates elements of DFM to a greater extent. The overall weight is kept low due to weight reduction features and design changes, particularly the cavities on the vertical walls and cavities in the bottom portion. Other improvements include better fillets and rounded edges to help with the casting process [30]. The geometry of the bottom of the Pitman frame was modified to accommodate a webbed-rib design to facilitate easier core removal and to give better strength to the casting. Improvements made to improve castability also had a significant impact on the stress values seen in Fig 8, while also reducing the weight of the casting. As shown in Table 4, these improvements increased the value of F by a considerable amount. Since Version 2 utilizes DFM to leverage manufacturing processes, $MS_3 = 0.06$ and $\Sigma MS = 0.06$, as shown in Table 4.

Version 3, shown in Fig 7, is a biomimetics-design based on the structure of the jaws of carnivores. Consequently, the measurements of several jaws of carnivores were taken into consideration [44]. The ratio of jaw length to jaw height was

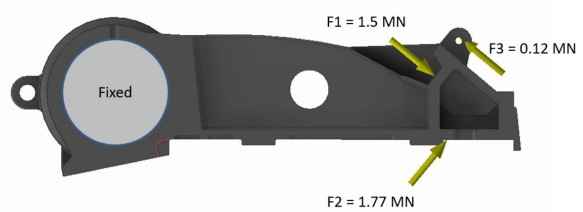

**Fig 6. Loads and constraints acting on the Pitman frame.** F1 is the reaction force due to the toggle plate, F2 is the crushing force and F3 is the spring load from the toggle mechanism.

V1 V2 V3

**Fig 7. Versions 1, 2 and 3 of the design of the Pitman frame.**

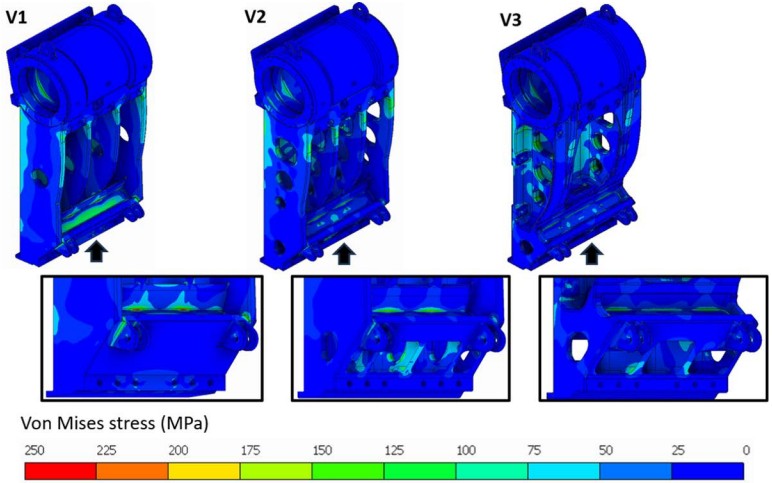

V1 V2 V3

Von Mises stress (MPa)

250 225 200 175 150 125 100 75 50 25 0

**Fig 8. Stress analysis of the Pitman frame – Rear view shown in insets.**

**Table 4. Summary of results for the Pitman frame design.**

| Version | 1 | 2 | 3 |
|---|---|---|---|
| Mass (Kg) | 2932 | 2764 | 2720 |
| Yield strength *(MPa)* | 276 | 276 | 276 |
| Peak Stress *(MPa)* | 281.4 | 186.2 | 164.1 |
| Max displacement *(mm)* | 2.081 | 1.876 | 1.63 |
| N (Peak stress/ Yield strength) | 0.98 | 1.48 | 1.68 |
| ΣMS | 0 | 0.06 | 0.08 |
| F | 0.98 | 1.54 | 1.76 |
| $F^N$ | $0.98^{0.98}$ | $1.54^{1.48}$ | $1.76^{1.68}$ |

plotted against the ratio of calculated bite force to jaw length. This number was close to 2 for animals that had the strongest bite force per unit length of their jaw. Also, this version of the Pitman frame has a ratio of 2.1 between the length up to the toggle seat and total height of the Pitman frame, which is ideal as per the observations on the measurements of animal jaws. Accordingly, a twin wall structure was implemented inspired by the shape of the lower jawbone. The side walls extend from the bearing location to the toggle seat, incorporating a beaded rib design due to its load-bearing capacity [30]. Although this design resulted in a slight decrease in weight of the casting, it improved the component's strength largely by reducing the peak stress value as seen in Table 4. Other small changes like rounding off the corners of cavities also reduced stresses in the load path, see Fig 8. As seen in Table 4, the resulting increase in the value of N, contributed to the increased value of F for this version of the design. Since Version 3 also uses the same design principles as Version 2, $MS_3 = 0.06$. Additionally, biomimetics contributes to material savings, giving $MS_5 = 0.02$. Therefore, $\Sigma MS = 0.06 + 0.02 = 0.08$, as shown in Table 4.

### Rear wall

The rear wall of a jaw crusher holds one of the ends with the other being held by the front wall. The rear wall also performs the function of conducting the crushing forces through the structure and into the foundation. The rear wall is also subjected to high levels of loading like the Pitman and thus the same material, i.e., ASTM A352, is used for this casting. Since the toggle acts as a fuse between the Pitman and the rear wall, the design load of the toggle plate is again taken as an input for the stress analysis of the rear wall casting. The toggle seat is accordingly loaded with *1.5 MN* of force. A separate analysis was done on the motor frame to determine the reaction forces on the bolts connecting the rear wall and this frame. These values were small compared to the *1.5 MN* toggle force and did not have a significant impact on the overall analysis. An additional force of *0.12 MN* corresponding to loading from the tension spring is added at the respective assembly point. The 3D models for the different versions were put though a stress analysis in Autodesk® Inventor® with the same loading and constraints.

Version 1, shown in Fig 9, is a simple design based on the available geometrical constraints. Cavities were introduced where necessary to reduce weight, but DFM was not fully utilized in this version. The results of the stress analysis, shown in Fig 10, reveal that the stress is concentrated near the bosses where the rear wall is assembled with the side walls. This can be easily addressed by increasing the root radius of the boss. However, the wall thickness of Version 1 is higher in some regions, and this leads to a rigid and heavy casting. Therefore, application of the *frugal design approach* entails mass removal where the stress is low.

Version 2, shown in Fig 9, exploits the casting process by introducing some degree of complexity in the shape and reducing unwanted material. In particular, a ribbed design with fillets [30] was adopted for effective usage of material. Cavities were also made in the casting including hollow, ribbed sections in the internal structure and weight saving holes

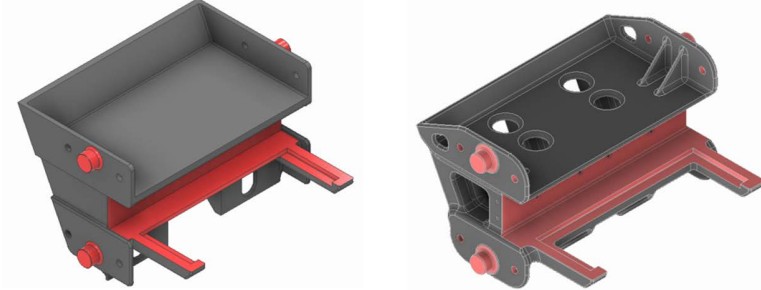

**Fig 9. Versions 1 and 2 of the design of rear wall.**

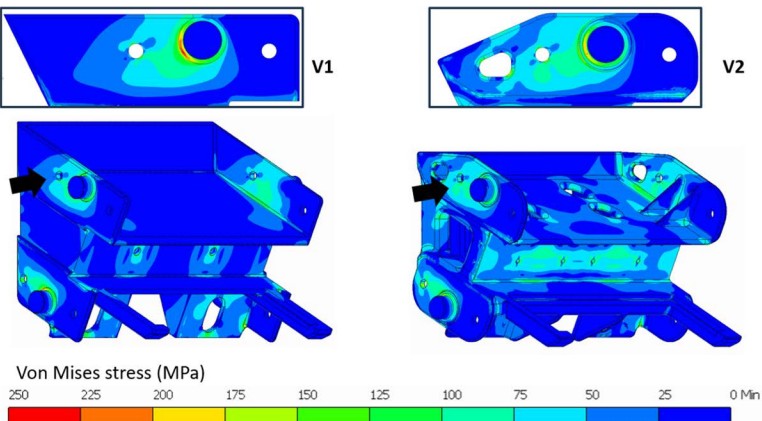

**Fig 10. Stress analysis of rear wall.** Front view with arrows indicates the insets of high stress areas.

at the top of the component. Material was removed wherever the component was not loaded. The general wall thickness of the casting was also maintained uniformly. The resulting structure was more stressed at the center than Version 1, but it was under an acceptable level, see Fig 10. The peak stress was much lower due to better distribution of stress along with fillets made around the mounting bosses. The large amount of savings in material and the lower peak stress contribute to a high F value, as seen in Table 5. Since Version 2 utilizes DFM to leverage manufacturing processes, $MS_3 = 0.3$ and $\Sigma MS = 0.3$, as shown in Table 5.

## Discussion

A summary of the iterative process of FDA through design refinement of the chosen components is shown in Fig 11. The three components, i.e., flywheel, the Pitman and the rear wall, were designed using the *frugal design approach* shown in Fig 3. In all three cases, the initial design had a *factor of safety* (N) lower than *1.5*. The theoretical model for the $F^N$ approach fixes the value of N at *1.5*. However, when dealing with complex structures, occurring in real time, such as those in this paper, it is difficult to start off or finish with an exact value of N. Therefore, all the components were initially designed with an N value lower than *1.5* in order to keep material usage at a minimum. The approach, based on the results of stress analyses, was to remove material from the low stress zones while tackling high-stress zones with design changes. In the process of design refinement, care was taken not to weaken the structure while removing the material or modifying the design. Such rigorous and careful execution of the design process yielded an increase in the *factor of safety* together with material savings. Overall, the next version of a design was selected based on material savings along

**Table 5. Summary of results for rear wall design.**

| Version | 1 | 2 |
|---|---|---|
| Mass *(Kg)* | 1209 | 841 |
| Yield strength *(MPa)* | 276 | 276 |
| Peak Stress *(MPa)* | 243.4 | 207.4 |
| N (Peak stress/ Yield strength) | 1.13 | 1.33 |
| $\Sigma MS$ | 0 | 0.3 |
| F | 1.13 | 1.63 |
| $F^N$ | $1.13^{1.13}$ | $1.63^{1.33}$ |

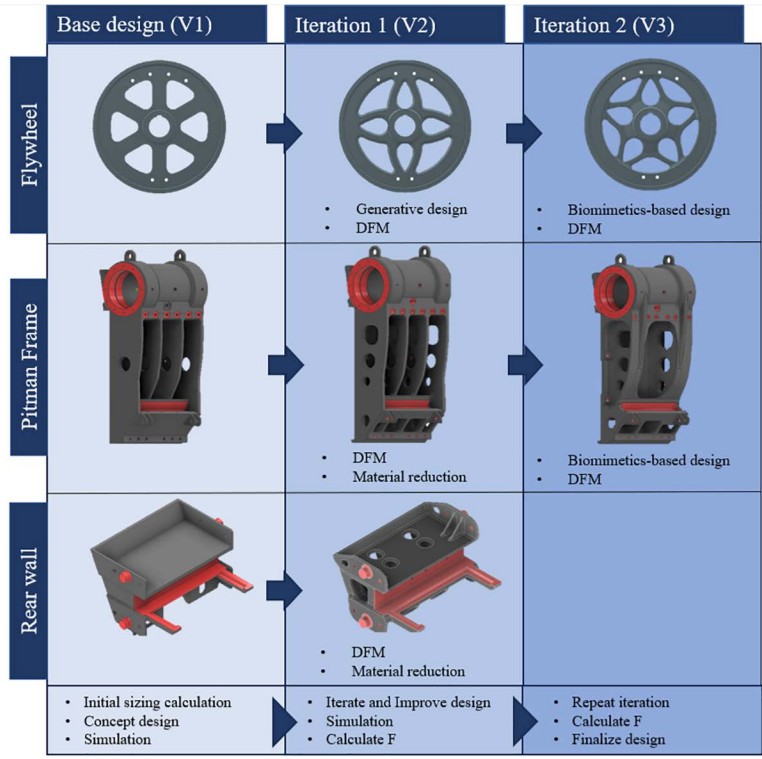

**Fig 11. Summary of design evolution, using Factor of Frugality, of relevant components of a jaw crusher.**

with the final N value being as close to *1.5* as possible. A comparison of the N, F and MS values of each design version is shown in Fig 12.

A brief analysis on how iterative changes in design affected the F value of each component now follows. As shown in Fig 12, the F value of the flywheel increased from *1.32* to *1.72* in the first iteration. This is attributed to the application of generative-design and DFM principles. In particular, engineering the arms of the flywheel to better handle bending reduced the stress concentration, thereby enabling material reduction through stronger design. Further refinement in the second iteration was achieved by following again, DFM rules specific to flywheel-manufacturing and, by using principles

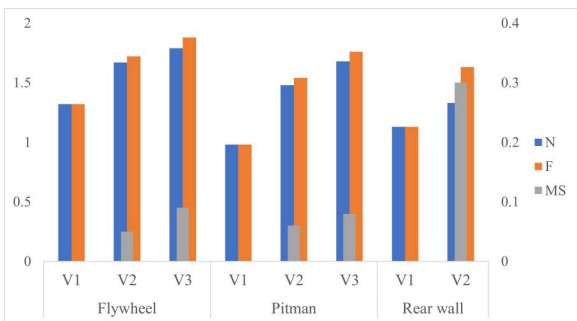

**Fig 12. Evolution in N, F and MS. This figure quantifies different design-versions for flywheel, the Pitman and rear wall.** N and F are plotted on the primary axis (Left) and MS is plotted on the secondary axis (Right).

of biomimetics for mimicking the structure of trees. The details of these design changes have been covered under the section on results. Consequently, additional branching and tapered profile of the flywheel-arms resulted in a stronger structure with lesser material, which has been captured in the final F value of *1.88*, see Fig 12.

Similarly, the first iteration of design for the Pitman frame, shown in Fig 12, increased the F value from *0.98* to *1.54*. This can be attributed to the change in design of the lower half of casting, wherein a webbed rib was adopted with larger fillets. This allowed saving on weight while improving strength. The second iteration of the Pitman sees an increase in F value from *1.54* to *1.76*. The focus in this iteration was on the structure of the frame itself. A twin-wall structure with a beaded rib was adopted for design. As outlined in the section on results, this design leverages the combined advantages of biomimetics and traditional engineering concepts. As for the rear wall, the casting went through a single iteration where the F value increased from *1.13* to *1.63*, see Fig 12. The design change involved removal of material in regions of low stress where the material was redundant. Such material reduction combined with larger fillets resulted in a lower peak stress and uniform stress distribution, as shown in Fig 10 and Table 5. The combination of material savings and stronger design contributed to the significant uptick in F-value of the rear wall.

The quantification of "frugality" in designs stemming from both lower *safety factors* and extraneous *material saving* schemes was accomplished using the *factor of frugality*. For the jaw crusher considered in this paper, $F^N$ is *$1.87^{1.79}$*, *$1.76^{1.68}$* and *$1.63^{1.33}$* for flywheel, the Pitman frame and rear wall, respectively. The flywheel has the highest F value among the three components. This is due to the higher N value of the flywheel that can be attributed to the adoption of biomimetics-based design and also the increase in endurance strength of the final version due to reduction in section thickness. At *1.68*, the final N value of the Pitman is very close to and above the recommended value of *1.5*, which is ideal. The N value of the rear wall comes close to the recommended value of *1.5* but is lower at *1.33*. The difference between the F and N values is highest in the rear wall, indicating a high degree of material savings, [18] due to the initial design being bulkier compared to the other components.

## Impact on cost

The components of the redesigned jaw crusher have saved *678 Kg* of material compared to the initial or baseline design. Accordingly, cost-benefit accrues from the use of lesser material and reduced expenditures associated with the casting process. As for the latter, a lighter casting leads to proportional savings through reduction of wastage. This is because the weight of runners and risers, i.e., gating system necessary for molding a casting, is proportional to the weight of the casting itself. Moreover, since the refined components do not change in size or in the processing and machining required for their fabrication, it can be assumed that change in other manufacturing costs is insignificant. Therefore, based on materials and energy, the total cost savings can be expressed as: [45]

$$S_{total} = S_{material} + S_{energy} \tag{3}$$

$$S_{material} = C_{metal} \times W_{metal} \times 1.1 \tag{4}$$

$$S_{energy} = C_{energy} \times E_{melting} \times W_{metal} \times 1.3 \tag{5}$$

Where,

$S_{total}$= Total cost saved
$S_{material}$= Material cost savings
$S_{energy}$= Energy cost savings
$C_{metal}$= Price of metal
$W_{metal}$= Weight of metal

$C_{energy}$ = Energy price

$E_{melting}$ = Energy required for melting

In the above formulations, the energy required for holding the melt has been ignored, assuming the melt is poured into the mold directly after melting. The factor of *1.1* in equation (4) accounts for any loss of the metal while *1.3* in equation (5) accounts for proportionate savings in risers and gates. The energy price is taken as *$0.08/kWh (6.9 INR/kWh)* [46] and the melting-energy is taken to be *0.58 kWh/Kg* [45].

As seen in Table 6, the total saving in costs based on equations (3) to (5) is estimated to be *$650.70* for the entire assembly. It is implicitly assumed that manufacturing has been carried out in a cost-effective way in India [47].

## Impact on performance

The *frugal design approach* was used in improving the design of critical components including flywheel, the Pitman and rear wall. The components were designed based on the significance of the toggle plate to the jaw crusher. The toggle plate thus formed the basis for applying loading to the components. This in turn determines the maximum size or the hardness of the material that can be crushed, as per Equation (2). Of the three components studied in this work, the rear wall has the least N value of 1.33. Accordingly, if the toggle plate and the shaft are designed to accommodate *33%* higher loads, i.e., designed with the same factor of safety, the size of the input material can be increased from *0.6 m* to *0.69 m*. This is based on using granite with a modulus of rupture of *25 MPa*. The throughput of the jaw crusher is directly proportional to the gap between the jaws of the crusher, and therefore the feed size [20]. For the same reduction ratio, the improved frugal design of the jaw crusher results in 15% higher throughput owing to accommodation of a higher feed size. Alternatively, the jaw crusher would theoretically be able to handle materials with a modulus of rupture of *33 MPa* for the same dimensions.

Although testing is required to validate these numbers, the design methodology, based on the factor of frugality, shows the ability to improve performance while also saving on material use. Therefore, the potential of this framework for the systematic design of advanced frugal products, such as the jaw crusher, has been demonstrated in this paper.

## Calculation of the factor of frugality for assemblies

Previous works on the *factor of frugality* [3,18,19] were focused on quantifying frugality for an individual component. However, a method for aggregating the *factor of frugality* of an assembly of frugal components has not been reported. This paper proposes quantification of the overall frugality of an assembly through a linear combination of individual *frugality factors* with the individual weights being dependent on the initial masses of the components. In doing so, heavier components contribute more to the overall frugality vis-à-vis lighter ones. Accordingly, the total value of F can be determined from the following equation.

**Table 6. Details of cost savings.**

| Component | Qty | Material | Price/ Kg* (USD) | Wt. saved (Kg) | Saving in material costs (USD) | Saving in energy costs (USD) | Total cost savings (USD) |
|---|---|---|---|---|---|---|---|
| Flywheel | 2 | Cast iron | 0.60 | 98 | 58.80 | 5.91 | 64.71 |
| Pitman | 1 | Alloy steel | 0.95 | 212 | 201.40 | 12.79 | 214.19 |
| Rear wall | 1 | Alloy steel | 0.95 | 368 | 349.60 | 22.20 | 371.80 |
| Total | | | | 678 | 609.80 | 40.90 | 650.70 |

*Price/Kg can be found in the SIAM commodity price data [48]. Rates have been converted at an exchange rate of 86 INR = 1 USD and rounded to two decimal places.

$$F_{ASSEMBLY} = \frac{\sum_{j=1}^{n} (M_1 \times F)_j}{\sum_{j=1}^{n} (M_1)_j}$$

(6)

where,

$n$ = number of parts in the assembly

$M_1$ = Mass of the initial model of the $j^{th}$ part

$F$ = Factor of frugality of the $j^{th}$ part

$F_{ASSEMBLY}$ = Total factor of frugality of the assembly

The flywheel, the Pitman and rear wall were selected and redesigned due to their major contribution to the weight of a jaw crusher. Therefore, the composite $F_{ASSEMBLY}$ for the jaw crusher will account for their actual F values. A value of *1.5* will be assigned to the remaining components listed in Table 1, as per the initial value recommended for the frugal design process [3]. This is assuming that these remaining components satisfy the basic functional requirements and are not put through the frugal design process. The calculation of $F_{ASSEMBLY}$ is listed in Table 7. Accordingly, $F_{ASSEMBLY}$ is determined, from Equation (6) and Table 7, to be *1.74* for the final, frugally engineered jaw crusher. In contrast, the $F_{ASSEMBLY}$ of the initial design of the jaw crusher is lower at *1.37.* Therefore, for ~*27%* increase in $F_{ASSEMBLY}$, the jaw crusher can accommodate *33%* higher loading and *15%* bigger feed.

## Conclusions

This paper presents an *advanced frugal* version of a jaw crusher by utilizing lesser material while providing better and robust performance. This has been made possible through a new *frugal design approach*, based on the *factor of frugality*, that systematically saves material while aiming for both quality-performance and robustness, at low cost.

- The design approach based on the *factor of frugality* yielded material savings in the range of *7% − 30%.* In all cases, material savings have been achieved between the iterations. It also increased the *factor of safety* by at least *18%* in the components studied. The *factor of safety* in the final iteration is slightly higher than the targeted value of *1.5*, thereby leaving room for further material savings through design optimization.

- Fixing the *safety factor* at *1.5* was difficult for complex parts. Hence the approach was tweaked by starting between *1* and *1.5.* Subsequent design iterations yielded a final *factor of safety* close to *1.5*, i.e., *1.25* to *1.75.*

- The *factor of frugality*-based design-approach yields significant material savings. This translates to significant cost savings, including expenditures from attendant energy consumption. The case for lower costs is further strengthened when components are frugally manufactured in emerging markets like India.

- The *factor of frugality* goes beyond the *classical factor of safety* in looking for more material savings while maintaining performance-quality at low-cost. Therefore, the *factor of frugality* is a modern version of the traditional *factor of safety* that will help designers engineer their parts/products for low-cost and low-resource in addition to safety through best performance.

**Table 7. Calculation of $F_{ASSEMBLY}$.**

| Name of the component | $M_1$ (Kg) | $F_{initial}$ | $F_{final}$ | Qty. | $M_1 * F_{initial}$ | $M_1 * F_{final}$ |
|---|---|---|---|---|---|---|
| Pitman | 2932 | 0.98 | 1.76 | 1 | 2873.36 | 5160.32 |
| Rear wall | 1209 | 1.13 | 1.63 | 1 | 1366.17 | 1970.67 |
| Flywheel | 627 | 1.32 | 1.87 | 2 | 1655.28 | 2344.98 |
| Other components | 4607 | 1.5 | 1.5 | – | 6910.50 | 6910.50 |
| Sum Total | 9375 | – | – | – | 12805.31 | 16386.47 |

## Notation

| Acronyms | |
|---|---|
| AFI | Advanced Frugal Innovation |
| DFF | Design For Frugality |
| FMI | Frugal Manufacturing Index |
| CAD | Computer-Aided Design |
| DFM | Design for Manufacturing |
| SDG | Sustainable Development Goals |

*Symbols*

| Symbol | Description | Units |
|---|---|---|
| $N$ | factor of safety | Dimensionless |
| $MS_i$ | Fraction of material savings using the $i^{th}$ material savings scheme | Dimensionless |
| $F^N$ | factor of frugality | Dimensionless |
| $F_{ASSEMBLY}$ | Assembly factor of frugality | Dimensionless |
| $\sigma$ | Modulus of Rupture | $MPa$ |
| $P$ | Force | $N$ |
| $L$ | Length | $m$ |
| $b$ | Breadth of the specimen | $m$ |
| $d$ | Depth of the specimen | $m$ |
| $I$ | Mass moment of inertia | $Kg\text{-}m^2$ |
| $S_{total}$ | Total cost saved | $INR$ |
| $S_{energy}$ | Energy cost savings | $INR$ |
| $S_{material}$ | Material cost savings | $INR$ |
| $C_{metal}$ | Price of metal | $INR/Kg$ |
| $W_{metal}$ | Weight of metal | $Kg$ |
| $C_{energy}$ | Energy price | $INR/kWh$ |
| $E_{melting}$ | Energy required for melting | $kWh/Kg$ |

## Author contributions

**Conceptualization:** Aditya Krishnan, Balkrishna C. Rao.

**Formal analysis:** Aditya Krishnan, Balkrishna C. Rao.

**Investigation:** Aditya Krishnan, Balkrishna C. Rao.

**Methodology:** Aditya Krishnan, Balkrishna C. Rao.

**Project administration:** Balkrishna C. Rao.

**Resources:** Aditya Krishnan, Balkrishna C. Rao.

**Software:** Aditya Krishnan.

**Supervision:** Balkrishna C. Rao.

**Validation:** Aditya Krishnan, Balkrishna C. Rao.

**Visualization:** Aditya Krishnan, Balkrishna C. Rao.

**Writing – original draft:** Aditya Krishnan, Balkrishna C. Rao.

**Writing – review & editing:** Aditya Krishnan, Balkrishna C. Rao.

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
