## [Decision Letter · Decision Letter 0]

PONE-D-25-09992Frugal engineering of a jaw crusher for extractive industriesPLOS ONE

Dear Dr. Rao,

Thank you for submitting your manuscript to PLOS ONE. After careful consideration, we feel that it has merit but does not fully meet PLOS ONE’s publication criteria as it currently stands. Therefore, we invite you to submit a revised version of the manuscript that addresses the points raised during the review process.

We look forward to receiving your revised manuscript.

Kind regards,

Gaurav Ashok Bhaduri

Academic Editor

PLOS ONE

Journal Requirements:

Additional Editor Comments (if provided):

Dear Authors

Thank you for submitting your manuscript to PLOS One. I do feel that the manuscript needs major revisions in order to be accepted for publication in PLOS One, based on the recommendations of the referees.

Please find the comments of the referees for your perusal and necessary amendments. Please resubmit the manuscript by 06/05/2025.

Kind Regards

Gaurav

Reviewers' comments:

Reviewer's Responses to Questions

**Comments to the Author**

1. Is the manuscript technically sound, and do the data support the conclusions?

Reviewer #1: Partly

Reviewer #2: Yes

2. Has the statistical analysis been performed appropriately and rigorously? 

Reviewer #1: Yes

Reviewer #2: Yes

3. Have the authors made all data underlying the findings in their manuscript fully available?

Reviewer #1: Yes

Reviewer #2: Yes

4. Is the manuscript presented in an intelligible fashion and written in standard English?

Reviewer #1: Yes

Reviewer #2: Yes

5. Review Comments to the Author

Reviewer #1: 1. The title “Frugal engineering of a jaw crusher for extractive industries” is too general. It is necessary to specify the unique aspect of the study, for example, mention “factor of frugality” or “bio-inspired design”.

2. Similarly, it is worth adding more specialized keywords.

3. The abstract gives a good overview of the study, but could be improved by adding quantitative results (for example, percentage weight reduction or increase in safety factor) for greater credibility. It is important to immediately indicate how much more “economical” the new design is in numerical terms.

4. The introduction well justifies the relevance of frugal engineering, but lacks a clearer statement of the problem that this article solves. What is the specific “disease” of current solutions that the authors want to “cure”? Also, it would be worthwhile to expand the literature review on frugal engineering in the mining industry to demonstrate the novelty of the approach.

5. “Use of the factor of frugality” – the very concept of “factor of frugality” requires a more detailed explanation. How is it calculated, what parameters are taken into account, why these particular ones? In formula (1), it is necessary to explain what “i” (index) is and what specific “material savings schemes” are implied.

6. The numerical modeling section lacks information on model validation. How is the adequacy of the finite element analysis model used confirmed? References to relevant standards, experimental data, or comparison with other known models are needed.

7. Version 1 (the basic design) is described as “simple”, but there is no detailed explanation of why this particular design was chosen as a starting point. Version 2 is created using the generative design method. However, there are few details about the parameters and constraints used in the generative design. Version 3 is bio-inspired, but the connection to specific natural objects is not obvious.

8. Tables 3, 4, and 5 present the results for each version of the components. However, there is no detailed analysis of these results. Why do some parameters improve while others deteriorate? What are the reasons for these changes? It is necessary to relate the results to the specific design features of each version.

9. It is important to show that the “factor of frugality” is not just an abstract value, but actually reflects resource savings and increased safety. What is the correlation between the “factor of frugality” and other performance indicators (e.g. production cost, service life, reliability)?

10. The presented cost savings calculation is too simplified. It does not take into account the costs of developing new designs, making prototypes, conducting tests, and possible changes in the technological process. Formulas (4) and (5) use coefficients 1.1 and 1.3 to account for metal losses and savings on sprues. It is necessary to justify the choice of these coefficients with references to literature or your own calculations.

11. All indicators are highly dependent on the material. It would be good to consider other materials and compare the indicators.

Reviewer #2: Although this paper represents a very interesting topic to the field and probably considered all processes, however it relies on granite as the crushed material while in nature the granite as raw material may be found mixed with other material particles such as iron. It’s recommended to take that into consideration.

6. PLOS authors have the option to publish the peer review history of their article (what does this mean? ). If published, this will include your full peer review and any attached files.

**Do you want your identity to be public for this peer review?** For information about this choice, including consent withdrawal, please see our Privacy Policy .

Reviewer #1: No

Reviewer #2: No

---

## [Author Response · Author response to Decision Letter 1]

5 May 2025

Frugal engineering of a jaw crusher using the factor-of-frugality, a modern version of the safety factor

At the outset we would like to thank the reviewers for their positive and valuable comments regarding this research effort. We also thank them for taking the time to point out meticulously issues pertaining to the content of this work. We have accordingly answered the various queries and revised the initial manuscript. Please find answers to the questions raised by the reviewers in the pages to follow. These useful comments have been used to revise the initial draft of our manuscript, which will be uploaded alongside these comments. Please note that the edited portions of the manuscript appear in red font in the revised manuscript (and which are underlined here). Also, some edits such as capitalizing certain words in the manuscript (shown in red-colored font) are not documented here for clarity and brevity.

REVIEWER 1

Comment 1: The title “Frugal engineering of a jaw crusher for extractive industries” is too general. It is necessary to specify the unique aspect of the study, for example, mention “factor of frugality” or “bio-inspired design”.

Response: The authors thank the reviewer for this suggestion.

Action Taken: Page 1, Line 1-2: Frugal engineering of a jaw crusher using the factor-of-frugality, a modern version of the safety factor

Comment 2: Similarly, it is worth adding more specialized keywords.

Response: The authors thank the reviewer for this suggestion and keyword listing has been changed.

Action Taken: Page 1, Line 22-23: Jaw crusher, Factor of frugality, Sustainability, Frugal engineering, Frugal design, Frugal manufacturing, Biomimetics, Advanced frugal product

Comment 3: The abstract gives a good overview of the study, but could be improved by adding quantitative results (for example, percentage weight reduction or increase in safety factor) for greater credibility. It is important to immediately indicate how much more “economical” the new design is in numerical terms.

Response: The authors thank the reviewer for this suggestion. And we have also added some minor content for a better reading of the abstract.

Action Taken: Page 1, Lines 15-19: Numbers indicated in the discussion and conclusion sections have been added in the abstract, as below.

Accordingly, factors of frugality of 1.871.79, 1.761.68 and 1.631.33 have been obtained for the flywheel, the Pitman and the rear wall, respectively, based on the frugal approach. Therefore, use of the new frugal design approach has resulted in material savings of 8%, 7% and 30% and increases in factor of safety of 35%, 71% and 18% in the flywheel, the Pitman and the rear wall, respectively, over their base values.

Minor content: Page 1, Lines 5-16:

Businesses are increasingly keen on going frugal due to increasing demand for sustainable and low-cost products that do not sacrifice quality. However, there is a dearth of tools for the systematic design and engineering of frugal products from scratch in the industry. Accordingly, a new approach has been applied in this paper for the formal design of a frugal jaw crusher for the mining industry. Consequently, this paper uses the factor of frugality (F of FS) which is a composite number that combines the safety factor (S) with fractions of material saved in various stages of product-development. In doing so, this work has iteratively applied the factor of frugality to the relevant components of a jaw crusher. And rigorous design procedures are adopted, for maintaining quality, due to the use of lower safety factors in making the product frugal. Contemporary concepts like generative design, design for manufacturing and biomimetics have been explored to achieve frugality in the relevant “bulky” components of a jaw crusher.

Comment 4: The introduction well justifies the relevance of frugal engineering, but lacks a clearer statement of the problem that this article solves. What is the specific “disease” of current solutions that the authors want to “cure”? Also, it would be worthwhile to expand the literature review on frugal engineering in the mining industry to demonstrate the novelty of the approach.

Response: The authors thank the reviewer for this suggestion. However, the authors would like to point out the following real-world problems that have been addressed in the paper:

1. Instead of quantifying the ‘frugality’ of a product purely using cost, a factor-of-frugality based approach is used, which captures the characteristics of going frugal, i.e., low-resource-consumption and better quality in addition to low-cost.

2. The focus of frugalization involving cost-reduction has also been shifted from the traditional avenues of labor and supply chain. Instead, frugal engineering principles have been applied at the design stage to directly influence material and cost savings while maintaining top quality. The application at design stage significantly helps in involving frugality in all stages of product development.

3. Up until now, the factor of frugality formula was developed to frugally engineer a single component of a system. A method to quantify the frugality of an entire assembly has been introduced in this paper.

In the introduction, it is also noted that this paper presents the first-ever instance of applying the factor-of-frugality based approach to an industrial product. Literature review of frugal products in the mining industry did not yield any results as products in the mining industry are typically characterized by bulky designs where robustness is more important than cost. Hence the absence of literature on frugal design from mining area which does not exist. In the interest of material savings and sustainability, a novel frugal design approach is adopted.

Action Taken: Accordingly, the issues that the paper addresses have been explicitly mentioned in the introduction with inclusion of previous content under point number 4.

Page 3, line 76-91:

Therefore, this paper addresses the novelty of applying frugal engineering principles to a jaw crusher by focusing on:

1. Frugal design is quantified using the factor of frugality. This allows the quantification of frugality in any product, including the jaw crusher, using a metric that hews with the three features of frugal engineering, i.e, low-cost, low-resource, and better quality.

2. The impact of the factor of frugality based design has been studied, using performance and cost as metrics.

3. The factor of frugality has been formulated for design of individual components of a given product. Consequently, a method of determining the overall factor of frugality of an assembly based on this formulation has been introduced in this effort.

4. This paper is the first instance of the methodical application of the factor of frugality to an actual industrial product. Accordingly, this paper presents the different versions of the product resulting from the iterative application of the factor of frugality while converging onto the final design. The advantages and limitations of the frugal methodology will become apparent in the process of designing this industrially relevant product.

Comment 5: “Use of the factor of frugality” – the very concept of “factor of frugality” requires a more detailed explanation. How is it calculated, what parameters are taken into account, why these particular ones? In formula (1), it is necessary to explain what “i” (index) is and what specific “material savings schemes” are implied.

Response: We thank the reviewer for this comment. Since this is a new factor with a new concept, we have provided details on pages 5-8 and Figures 2 &3. We have added new content to make this clear. We feel that the existing and added content on this new factor makes this concept clear. But the interested reader can look, if needed, for more details into Rao’s works that are referenced in the paper.

Action Taken: Accordingly, the following lines have been added.

Page 5, Lines 120-123 : The factor of frugality is a new metric for design and, engineering in general, that achieves best functionality under resource-and-cost constraints by subsuming the classical safety factor while also focusing on individual stages in product development. It is a modern version of the factor of safety that focuses on safety, resource, cost and quality in design.

Page 5, Lines 124-126: The factor of frugality approach is initiated by fixing N at a low value of 1.5, as seen in Fig 2 [3]. The low value of N is arbitrary in that it should be lowest possible value commensurate with the current body of knowledge in a given area of engineering.

Page 6, Lines 129-138: Fig 2 brings out the working of the factor of frugality, i.e, F of FN, for a shaft which is a workhorse of many engineering applications. As seen in Fig 2, a low value of N, at 1.5, is fixed throughout the frugal product development process. Such a low value of N generally results in lower material consumption and subsequent stages of product development are examined for more material savings in addition to that for this low value of N. Consequently, the material saved (MS) parameters account for weight of material saved in the design, materials and manufacturing stages with focus on salvaging or recycling. Therefore, low-resource consumption and, hence lower costs, are made possible from both low N and individual stages of product development. The quality is maintained at the highest level by hewing to the rigor of most accurate design and engineering principles.

Page 6, Line 141: However, some changes with this idealized setting of the factor of frugality are in order.

Comment 6: The numerical modeling section lacks information on model validation. How is the adequacy of the finite element analysis model used confirmed? References to relevant standards, experimental data, or comparison with other known models are needed.

Response: Page 9, Table 2 furnishes details about the type of mesh used and convergence criteria used. The h-refinement of the mesh is stopped once max stress deviation is within 5% of that of the previous iteration. This is done so that a mesh independent result is achieved. This gives sufficient confidence for the convergence of the result. Furthermore, all analyses are done within the Hookean domain, with none of the analyses exceeding the yield strength of the material. This has been specified in the paper.

We understand the reviewer’s concern on model validation. This is the first part of a study that aims to build and use such a frugal jaw crusher. We hope to publish these results when they become available.

Action Taken: Some minor modification was carried out.

Page 8, Line 182: …domain since plastic flow is detrimental…

Comment 7: Version 1 (the basic design) is described as “simple”, but there is no detailed explanation of why this particular design was chosen as a starting point. Version 2 is created using the generative design method. However, there are few details about the parameters and constraints used in the generative design. Version 3 is bio-inspired, but the connection to specific natural objects is not obvious.

Response: We thank the reviewer for this comment. The base model (Version 1) is a concept that is made in CAD based on machines observed in the field. This base design is chosen as a starting point because these designs have a low factor of safety, less than or equal to 1.5, as recommended by the frugal design methodology specified in Rao’s works. Subsequent design iterations aim to reduce material without decreasing this factor of safety. In Version 2 of the flywheel, which employs generative design, it is mentioned that symmetry constraints were applied on mutually perpendicular planes passing through the axis of the flywheel (Page 11, line 249-250). This results in a symmetric design. In all three versions, a torque and a belt tension force are also applied as loads on the rim of the flywheel (Page 10, line 225). These are the constraints and boundary constraints that are required to simulate loads on a flywheel. As for the connection to biomimetics, the connection to trees and branching etc have been briefly explained and the interested reader can refer [41,42] under Version 3 on page 12. We believe that relevant biomimetic features have been briefly explained here.

Comment 8: Tables 3, 4, and 5 present the results for each version of the components. However, there is no detailed analysis of these results. Why do some parameters improve while others deteriorate? What are the reasons for these changes? It is necessary to relate the results to the specific design features of each version.

Response: We thank the reviewer for this comment. Tables 3, 4 and 5 are meant to be observations. An analysis of these results can be found in the ‘Discussion’ section. The changes between each subsequent iteration of the design is necessitated by the frugal design methodology shown in Figure 3. The details of these changes with reference to the numbers shown in the tables can be found in the ‘Results’ section, refer Page 9 – 17. However, we have added two paras to account for this comment as follows.

Action Taken: Page 19 -20, lines 397 – 420:

A brief analysis on how iterative changes in design affected the F value of each component now follows. As shown in Fig. 12, the F value of the flywheel increased from 1.32 to 1.72 in the first iteration. This is attributed to the application of generative-design and DFM principles. In particular, engineering the arms of the flywheel to better handle bending reduced the stress concentration, thereby enabling material reduction through stronger design. Further refinement in the second iteration was achieved by following again, DFM rules specific to flywheel-manufacturing and, by using principles of biomimetics for mimicking the structure of trees. The details of these design changes have been covered under the section on results. Consequently, additional branching and tapered profile of the flywheel-arms resulted in a stronger structure with lesser material, which has been captured in the final F value of 1.88, see Fig. 12.

Similarly, the first iteration of design for the Pitman frame, shown in Fig. 12, increased the F value from 0.98 to 1.54. This can be attributed to the change in design of the lower half of casting, wherein a webbed rib was adopted with larger fillets. This allowed saving on weight while improving strength. The second iteration of the Pitman sees an increase in F value from 1.54 to 1.76. The focus in this iteration was on the structure of the frame itself. A twin-wall structure with a beaded rib was adopted for design. As outlined in the section on results, this design leverages the combined advantages of biomimetics and traditional engineering concepts. As for the rear wall, the casting went through a single iteration where the F value increased from 1.13 to 1.63, see Fig. 12. The design change involved removal of material in regions of low stress where the material was redundant. Such material reduction combined with larger fillets resulted in a lower peak stress and uniform stress distribution, as shown in Fig. 10 and Table 5. The combination of material savings and stronger design contributed to the significant uptick in F-value of the rear wall.

Comment 9: It is important to show that the “factor of frugality” is not just an abstract value, but actually reflects resource savings and increased safety. What is the correlation between the “factor of frugality” and other performance indicators (e.g. production cost, service life, reliability)?

Response: We thank the reviewer for this comment. We have explained these features in bringing out the concept underlying the Factor of Frugality on pages 5 to 8. We have also added new content to explain the theoretical underpinnings of this new design-metric that captures both the resource savings and safety of a frugally engineered product while keenly focusing on cost and top quality. This has also been explained in detail in Rao’s works [2,3,12,18,19, 28].

As for the application to an industry grade jaw crusher, an increase in factor of safety from design iteration

---

## [Decision Letter · Decision Letter 1]

Frugal engineering of a jaw crusher using the factor-of-frugality, a modern version of the safety-factor

PONE-D-25-09992R1

Dear Dr. Rao,

We’re pleased to inform you that your manuscript has been judged scientifically suitable for publication and will be formally accepted for publication once it meets all outstanding technical requirements.

Kind regards,

sunny narayan

Academic Editor

PLOS ONE

Additional Editor Comments (optional):

Reviewers' comments:

Reviewer's Responses to Questions

**Comments to the Author**

1. If the authors have adequately addressed your comments raised in a previous round of review and you feel that this manuscript is now acceptable for publication, you may indicate that here to bypass the “Comments to the Author” section, enter your conflict of interest statement in the “Confidential to Editor” section, and submit your "Accept" recommendation.

Reviewer #2: All comments have been addressed

2. Is the manuscript technically sound, and do the data support the conclusions?

Reviewer #2: Yes

3. Has the statistical analysis been performed appropriately and rigorously? 

Reviewer #2: Yes

4. Have the authors made all data underlying the findings in their manuscript fully available?

Reviewer #2: Yes

5. Is the manuscript presented in an intelligible fashion and written in standard English?

Reviewer #2: Yes

6. Review Comments to the Author

Reviewer #2: thank you for your response, but it is very important point and may effect the whole process in sites so I think you might take this point into consideration in future research.

7. PLOS authors have the option to publish the peer review history of their article (what does this mean? ). If published, this will include your full peer review and any attached files.

**Do you want your identity to be public for this peer review?** For information about this choice, including consent withdrawal, please see our Privacy Policy .

Reviewer #2: No

---

## [Editor Report · Acceptance letter]

PONE-D-25-09992R1

PLOS ONE

Dear Dr. Rao,

I'm pleased to inform you that your manuscript has been deemed suitable for publication in PLOS ONE. Congratulations! Your manuscript is now being handed over to our production team.

Kind regards,

on behalf of

Dr. PLOS Manuscript Reassignment

Staff Editor

PLOS ONE